# Analysis of Pros and Cons in Using [^68^Ga]Ga-PSMA-11 and [^18^F]PSMA-1007: Production, Costs, and PET/CT Applications in Patients with Prostate Cancer

**DOI:** 10.3390/molecules27123862

**Published:** 2022-06-16

**Authors:** Costantina Maisto, Michela Aurilio, Anna Morisco, Roberta de Marino, Monica Josefa Buonanno Recchimuzzo, Luciano Carideo, Laura D’Ambrosio, Francesca Di Gennaro, Aureliana Esposito, Paolo Gaballo, Valentina Pirozzi Palmese, Valentina Porfidia, Marco Raddi, Alfredo Rossi, Elisabetta Squame, Secondo Lastoria

**Affiliations:** 1Nuclear Medicine Division, Istituto Nazionale Tumori-IRCCS-Fondazione G. Pascale, 80127 Napoli, Italy; c.maisto@istitutotumori.na.it (C.M.); a.morisco@istitutotumori.na.it (A.M.); roberta.demarino@istitutotumori.na.it (R.d.M.); monica.buonanno@istitutotumori.na.it (M.J.B.R.); luciano.carideo@istitutotumori.na.it (L.C.); l.dambrosio@istitutotumori.na.it (L.D.); f.digennaro@istitutotumori.na.it (F.D.G.); aureliana.esposito@istitutotumori.na.it (A.E.); p.gaballo@istitutotumori.na.it (P.G.); valentina.pirozzipalmese@istitutotumori.na.it (V.P.P.); valentina.porfidia@istitutotumori.na.it (V.P.); marco.raddi@istitutotumori.na.it (M.R.); alfredo.rossi@istitutotumori.na.it (A.R.); e.squame@istitutotumori.na.it (E.S.); 2Hospital Pharmacy Department, ASL 1 “Ospedale del Mare” Hospital, 80127 Napoli, Italy; michela.aurilio@aslnapoli1centro.it

**Keywords:** PCa, [^68^Ga]Ga-PSMA-11, [^18^F]PSMA-1007, PSMA, [^68^Ge]/[^68^Ga] generator, cyclotron RCY, RCP

## Abstract

The aim of this work is to compare [^68^Ga]Ga-PSMA-11 and [^18^F]PSMA-1007 PET/CT as imaging agents in patients with prostate cancer (PCa). Comparisons were made by evaluating times and costs of the radiolabeling process, imaging features including pharmacokinetics, and impact on patient management. The analysis of advantages and drawbacks of both radioligands might help to make a better choice based on firm data. For [^68^Ga]Ga-PSMA-11, the radiochemical yield (RCY) using a low starting activity (**L**, average activity of 596.55 ± 37.97 MBq) was of 80.98 ± 0.05%, while using a high one (**H**, average activity of 1436.27 ± 68.68 MBq), the RCY was 71.48 ± 0.04%. Thus, increased starting activities of [^68^Ga]-chloride negatively influenced the RCY. A similar scenario occurred for [^18^F]PSMA-1007. The rate of detection of PCa lesions by Positron Emission Tomography/Computed Tomography (PET/CT) was similar for both radioligands, while their distribution in normal organs significantly differed. Furthermore, similar patterns of biodistribution were found among [^18^F]PSMA-1007, [^68^Ga]Ga-PSMA-11, and [^177^Lu]Lu-PSMA-617, the most used agent for RLT. Moreover, the analysis of economical aspects for each single batch of production corrected for the number of allowed PET/CT examinations suggested major advantages of [^18^F]PSMA-1007 compared with [^68^Ga]Ga-PSMA-11. Data from this study should support the proper choice in the selection of the PSMA PET radioligand to use on the basis of the cases to study.

## 1. Introduction

Prostate adenocarcinoma (PCa) is the second most commonly diagnosed cancer among males and the fifth cause of death worldwide related to disease progression [1]. Advanced prostate cancer represents the ultimate challenge in terms of reducing its specific mortality. Accurate diagnosis and efficient staging of recurrences in metastatic castration-resistant prostate cancer (mCRPC) are greatly required.

In the last years, it has been demonstrated that prostate-specific membrane antigen (PSMA) plays a pivotal role in the detection, at diagnosis, of primary PCa as well as of recurrent and metastatic sites of disease, being an optimal target highly overexpressed in poorly differentiated, metastatic, and hormone-refractory PCa [2,3]. In fact, PSMA PET/CT has been recently included in international guidelines for imaging in biochemical recurrence (BCR) [4]. Furthermore, there is a high demand for monitoring patients with relapsed or metastasized prostate cancer because PSMA-specific PET tracers are not negatively affected by androgen deprivation therapy (ADT) once metastases became castration resistant. Conversely, for castration-resistant patients, the impact of ADT on PSMA expression in hormone-sensitive prostate cancer (HSPC) remains unclear and is under investigation. Several radiolabeled PSMA-inhibitors have been developed and, among these, two classes of probes gained particular relevance for clinical use as PET/CT radiopharmaceuticals in PCa: those labeled with gallium-68 [^68^Ga] including PSMA-11, PSMA I&T, PSMA-617, etc., and those labeled with Fluoride-18 [^18^F] including PSMA-1007, DCFPyL, and JK-PSMA-7, etc. All of these synthetic peptides, independently from the radionuclide used for PET imaging, are characterized by high affinities for PSMA, although different, and ensure excellent performances in terms of the detection rate and diagnostic accuracy. Different patterns of biodistribution and variable levels of uptake in normal organs as well as pharmacokinetic properties were found. For such characteristics, these agents are in Pharmacopeia or under evaluation for authorization by regulatory agencies worldwide and they rapidly gained great relevance in the management of PCa patients. Thus, the knowledge of pros and cons of ^68^Ga-labeled and ^18^F-labeled-PSMA is mandatory for the proper choice. In this setting, we have analyzed our data of the past years at NCI of Napoli, by reviewing all of the parameters involved in the preparation and use of [^68^Ga]Ga-PSMA-11 and [^18^F]-PSMA-1007. In detail, we considered the radiochemical yields, the imaging performances, and the impact of PET findings in the management of patients with PCa, including the selection of eligible patients for radioligand therapy (RLT) [5]. Finally, we evaluated the cost of production, the amount of the final product along with the number of allowed PET/CT exams. To evaluate the costs of production of [^68^Ga]Ga-PSMA-11 and [^18^F]PSMA-1007 in a nuclear medicine center with its own cyclotron/radiopharmacy facility, we adopted the same procedure previously published for [^18^F]PSMA-1007 [6]. The knowledge obtained from all of these data will support the choice of the best option to adopt in a nuclear medicine center. 

## 2. Results

### 2.1. Production/Labeling

Two starting activities of [^68^Ga]Gallium chloride, low (L) (596.55 ± 37.97 MBq) and high (H) (1436.27 ± 68.68 MBq), were used for the labeling of PSMA-11 (20 productions, 10 for each activity range). This choice was driven by the eluted radioactivity in two different periods of [^68^Ge]/[^68^Ga] generator shelf life. In detail, H activities correspond to the first seven weeks of generator life from the calibration date, and L activities correspond to the last seven weeks of generator life.

For all produced batches, the quality controls, performed on the final products, were in compliance with the acceptance criteria described by Ph. Eur. XII.

The values of radiochemical purity (RCP), evaluated by radio-HPLC and radio-iTLC, were ≥99%. Residual ethanol was ≤10% *v*/*v*. 

Radionuclide purity, assessed by γ-spectrometry, ranged from 490 to 531KeV, while the half-life measurement was between 62 and 74 min, both in the normal range. 

The endotoxin value, determined by the Limulus Amebocyte Lysate test (LAL), was ≤2.5 EU/mL.

The RCP at the end of synthesis (EOS) and the related RCY are summarized in Table 1. The average value of RCY for L activity, not corrected for decay, was 80.98 ± 0.05%, decreasing to 71.48 ± 0.04% for H activity. The RCP was unaffected by the initial [^68^Ga]Gallium chloride activity, resulting in 99.91% (99.75–100%) for L and 99.96% (99.81–100%) for H activities.

The costs for each production of [^68^Ga]Ga-PSMA-11, besides the starting activity, was approximately 1.830€; ranging from 66€/37 MBq to 140€/37 MBq for high and low activities of [^68^Ga]Gallium chloride, respectively. These differences in terms of costs per 37 MBq were obtained by considering the number of allowed PET/CT exams: two PET/CT exams for L activities and up to five for H activities.

The labeling procedure, quality controls, and cost of production for preparing [^18^F]PSMA-1007 in our institution have been analyzed and published [6]. Considering a starting activity of about 90 GBq of [^18^F]Fluoride, a final activity of 45–50 GBq of [^18^F]PSMA-1007 was obtained, allowing 25–30 PET/CT exams per day. The costs for the production of [^18^F]PSMA-1007 was approximately 5.450€ (4.31€/37 MBq), as shown in Table 2.

### 2.2. PET/CT Imaging

[^18^F]PSMA-1007 and [^68^Ga]Ga-PSMA-11 have different patterns of uptake in normal organs, as shown in Figure 1A,B. The “physiological”, hepatic, uptake is higher for [^18^F]PSMA-1007 than for [^68^Ga]Ga-PSMA-11. Increased hepatic extraction of [^18^F]PSMA-1007 causes increased excretion throughout the intestine, and often prolonged retention within the gall bladder is observed. This finding is related to the higher lipophilicity of fluorinated radioligand. Such biodistribution might interfere with the detection of the involved abdominal lymph nodes. 

Increased uptake and elevated excretion via the urinary system are prevalent for [^68^Ga]Ga-PSMA-11. The prolonged and persistent presence of radioactive urine in the bladder might musk recurrences in the prostate and/or pelvic lymph nodes. 

Nevertheless, the detection of PCa deposits, in our experience, was not significantly affected by the behavior of the two radioligands in normal organs/tissues. In the wide majority of cases, both radioligands depicted the same lesions with different levels of uptake as shown in Figure 1. This patient had an initial PET/CT exam with [^18^F]PSMA-1007, performed before RLT with [^177^Lu]Lu-PSMA-617, and the radioligand was concentrated within skeletal and lymph nodal metastases. After the first two cycles of RLT, the patient was reevaluated by PET/CT with [^68^Ga]Ga-PSMA-11, that documented the same lesions although with different uptake.

PET/CT images of both radioligands were also compared with planar, whole-body SPECT/CT images, acquired for dosimetric purposes 96 h after the first cycle (7.4 GBq of [^177^Lu]Lu-PSMA-617). All of the known lesions were seen by the three different PSMA-radioligands. Such a finding indicates that the selection of candidates for RLT may be reliable, independently from the used PET-PSMA radioligand.

## 3. Discussion

In current clinical practice, PSMA-PET/CT is gaining a key role in the management of patients with PCa. It has been shown to be effective in the diagnosis, follow-up, namely, for patients with biochemical recurrence, and in the selection of patients to submit to RLT [7].

According to our experience with more than 2000 patients using either ^68^Ga-labeled or ^18^F-labeled PSMA, we have tried to summarize the advantages and the limitations of the two PET radioligands and their application in the clinical routine at our site. 

The wide majority of published papers used [^68^Ga]Ga-PSMA-11 and emphasized, as pro, the elevated diagnostic sensitivity (greater than 90%), and as a con the relatively low amounts of radioligand available at the end of each synthesis [8]. In fact, the amount of ^68^Ga-labeled compounds in general, and of PSMA in this specific case, is always defined by the activity eluted by the [^68^Ge]/[^68^Ga] generator, that ranges between 370 and 1110 MBq.

The RCY was affected by the activities of [^68^Ga]Gallium chloride, decreasing by about 10% for higher activities (1300 MBq–1500 MBq), usually obtained in the first seven weeks of generator life from the calibration date. 

In the near future, the production of [^68^Ga]Gallium chloride by cyclotron and a liquid target will likely improve the limited amounts of injectable radioligands by starting with activities 20–30 fold greater than those obtained with a generator [9,10]. RCY should be carefully evaluated in this new scenario. Previous reports have shown that larger amounts of [^68^Ga]Ga-PSMA-11 were obtained using cyclotron and solid targets. Activity as high as 72 GBq at EOS was obtained, with no evidence of negative effects on the quality of the final product (i.e., colloid or unreacted [^68^Ga]Gallium chloride). In terms of patient doses, this activity will allow 12–15 studies to be performed in a center with two PET cameras [11,12].

Conversely, the limit of low starting activity does not apply to the production of fluorinated PSMA inhibitors. The starting activity range of [^18^F]Fluoride plays a key role in the final yield for [^18^F]PSMA-1007; thus, the optimization of the variables influencing the yields of the synthesis is mandatory as we have previously demonstrated [6]. In fact, using different starting activities of [^18^F]Fluoride: **low** (55.91 ± 6.69 GBq), **medium** (89.06 ± 4.02 GBq), and **high** (162.38 ± 6.46 GBq), we found that the medium one was the most compliant for clinical needs, ensuring the best match of RCY, costs, and number of performed PET/CT exams [6].

For both radioligands, the measured RCP was ≥99% in all batches and it was unaffected by the different starting activities. 

Moreover, we considered the production costs of the two radioligands (including cyclotron beam/generator elution, ligand, disposables for dispensation, reagents, personnel and maintenance, and radioactive waste dismission including the wasting of disposables used for the synthesis processes). According to the actual volume of PSMA PET/CT exams performed in our institution, which ranges between 25–30 exams on a weekly basis, the costs of production for [^68^Ga]Ga-PSMA-11 will be much higher than those for [^18^F]PSMA-1007. They will range from 9.155€ to 18.310€ according to the [^68^Ge]/[^68^Ga] generator shelf life and the number of required syntheses (5 or 6) to prepare 25–30 doses. All of the required syntheses (5–6) could not be performed in one day because the [^68^Ge]/[^68^Ga] generator needs time to regenerate after elution [13]. As a reminder, according to our analysis, the cost of a [^18^F]PSMA-1007 production to perform the same number of PET/CT exams is approximatively 5.450€.

From a clinical point of view, the different chemical properties of the two diagnostic radioligands somehow influence their biodistribution; the timing of PET/CT acquisition does not significantly affect the diagnostic accuracy and their use in the various clinical scenarios of PCa patients, including the definition of eligibility for RLT.

Our experience indicated that the highest uptake within lesions was observed by [^18^F]-PSMA-1007 while the overall rate of detection was not significantly affected by the PSMA probe. The highest uptake of [^18^F]-PSMA-1007 may be explained by the highest injected activity (up to 3-fold) of [^68^Ga]Ga-PSMA-11 and the relatively high number of emitted positrons vs. [^68^Ga]Ga-PSMA-11, as shown in Figure 1.

At least two significant differences were detectable on PET images and were strictly related to the pharmacokinetic properties of these two radioligands as shown in Figure 2. A greater physiological hepatic retention occurs when using [^18^F]PSMA-1007, and it is related to the highest lipophilicity [14,15]. In kidneys and the urinary tract, [^68^Ga]Ga-PSMA-11 accumulates highly because of its greater hydrophilicity. Among the differences between the two radiopharmaceuticals, a greater [^18^F]PSMA-1007 uptake in the skeleton is frequently observed, which according to the extensive and prolonged quality controls we have performed in our experience, is not related to free [^18^F]Fluorine. The nature of the isotope may be a possible explanation: the lower positron energy and the higher rate of photon emission (photon flux density) of [^18^F]Fluorine compared to [^68^Ga]Gallium contribute to the detection of more positive benign lesions in the skeleton, related to increased osteoblastic activity (i.e., osteoarthritis, degenerative changes, fractures, etc.) [16,17]. 

The analysis of images might be in favor of fluorinated PSMA because its positron energy emission (0.65 MeV) that is lower than that of [^68^Ga]Gallium (1.90 MeV) enables a better spatial resolution on PET/CT images (Figure 1) and a lower radiation burden [13,14]. Furthermore, the positron yield of [^68^Ga]Gallium is lower than that of [^18^F]Fluoride (89.14% vs. 96.86%), increasing image noise by possible prompt gamma contamination in PET data and negatively impacting the detection sensitivity [14].

The anatomical sites of “normal” uptake are the same for both diagnostic PSMA ligands, and the different contrast on PET/CT images is due to the density of β^+^ generated by the decay, which is major for [^18^F]Fluoride [14,18].

A higher uptake of [^18^F]PSMA-1007 than [^68^Ga]Ga-PSMA-11 within recurrent metastases has been reported as well as a biodistribution similar to PSMA-617, currently used for the RLT of mCRPC [19]. Despite the differences in the structure of the molecules, in the radiolabeling strategy (a prosthetic rather than a chelator group), PSMA-1007 and PSMA-617 show reduced kidney uptake compared with PSMA-11. Thus, [^18^F]PSMA-1007 and [^177^Lu]Lu-PSMA-617 seem to be a perfect theragnostic tandem [20] as confirmed in our experience.

## 4. Materials and Methods

Radiosyntheses and quality controls of [^18^F]PSMA-1007 were performed as previously described [5].

### 4.1. Radiosynthesis of [^68^Ga]Ga-PSMA-11

[^68^Ga]Gallium was obtained by a ^68^Ge/^68^Ga generator (0.74–1.85 GBq) (Galliapharm, Eckert-Ziegler, Berlin, Germany). Radiosyntheses were performed using an All In One mini automated synthesizer (Trasis, Ans, Belgium) and cassettes without pre-purification after generator elution. The reagents kit (ABX, Radeberg, Germany) included PSMA-11 (GMP grade), acetate buffer (0.7 M), HCl (0.1 M), ethanol, and a saline bag.

Radiochemical yield was evaluated by radio-high performance liquid chromatography (HPLC) analyses that were carried out using an LC 20AD Pump with a SPD-20AV UV/VIS detector (Shimadzu, Kyoto, Japan) equipped with a GABI radiometric detector (Raytest, Elysia, Straubenhardt, Germany). A 5 µm C18 300 Å, 250 × 4.6 mm column (Jupiter®, Phenomenex, Bologna, Italy) was used with a flow rate of 1 mL/min and the following gradient (acetonitrile 0.1% Trifluoroacetic acid (TFA) as solvent A, water 0.1% TFA as solvent B): 100% A for 5 min, A 25% and B 75% for 3 min, the same gradient for 4 min, and then 100% B for 3 min. The UV/VIS detector was set at 220 nm and 254 nm.

Thin-layer chromatography (TLC) was carried out using 1 M ammonium acetate/methanol (1:1) as the mobile phase and iTLC-SG glass microfiber chromatography paper impregnated with silica gel (Agilent Technologies, Santa Clara, CA, USA) as the stationary phase. TLCs were analyzed by a storage phosphor system (Cyclon Plus, Perkin Elmer, Oxford, UK).

Residual ethanol was quantified by a gas-chromatography system (GC 2010 Plus, Shimadzu, Japan) equipped with a flame-ionization detector (FID) and a capillary column (Elite-1301, 6% cyanopropylphenyl 94% dimethyl polysiloxane, L 30 m, ID 0.53; Perkin Elmer, UK). The temperature for the split was 240 °C and 280 °C for the FID. 

Radionuclidic purity was determined using a multi-channel analyzer (Mucha Star, Raytest, Elysia, Germany) and by half-life measurement with a dose calibrator (Atomlab 500, Biodex, New York, NY, USA). 

A radiotracer was also tested for bacterial endotoxin through the kinetic chromogenic Limulus Amebocyte Lysate (LAL) method (Endosafe Nexgen-PTS, Charles River, Wilmington, MA, USA). 

All quality controls of the final products followed the acceptance criteria described in Ph. Eur. XII.

### 4.2. Imaging Protocols

PET/CT images were performed on a time-of-flight PET/CT digital scanner (Discovery MI, GE Healthcare, Waukesha, WI, USA) with a 25 cm axial field of view. A low-dose CT scan (120 kV; automated current modulation; 0.98 pitch; 3.75 mm slice thickness; 0.5-s rotation time) from the vertex of the skull to the mid-thighs was used for anatomic localization and attenuation correction. PET scanning followed at 2 min/bed position with a 27-slice overlap. Images were reconstructed using ordered-subset expectation maximization (OSEM, four iterations, eight subsets, cutoff 6 mm, 256 × 256 matrix) applying all appropriate corrections for dead time, randoms, scatter, coincidence, and detector normalization.

Post therapy images were acquired on a SPECT/CT Discovery NM/CT 670 system (GE Healthcare) equipped with MEGP (medium energy general purpose) collimators and 3/8′′ NaI(Tl) crystal thickness. A 20% energy window centered on the 208 keV photopeak and a 10% scatter correction window centered on 178 keV were applied. Whole-body images were acquired in continuous mode (15 cm/min), with a 1.0 zoom, 2.21 mm pixel size, and an automatic body contour.

## 5. Conclusions

[^18^F]PSMA-1007 is in many ways an ideal radiotracer for PCa imaging; it can be produced with purity and in high yield and it can be transported to sites that do not have cyclotron and radiopharmacy facilities.

[^68^Ga]Ga-PSMA-11 can also be produced in high purity but in much lower yield than [^18^F]PSMA-1007 unless it is produced by a cyclotron; moreover, [^68^Ga]Ga-PSMA-11 has to be produced on site due the short half-life and low yield. 

The costs of [^68^Ga]Ga-PSMA-11 production are the same for both starting activity ranges. However, comparing the two diagnostic radiopharmaceuticals, [^18^F]PSMA-1007 is the most compliant one, according to the aspects previously evaluated. 

From a clinical point of view, our experience suggests that [^18^F]PSMA-1007 is characterized by a better specificity in detecting recurrences and a similar behavior to the therapeutical analogue [^177^Lu]Lu-PSMA-617. This makes the radioligand a better probe for patients eligible for RLT.

To optimize the production of [^68^Ga]Gallium chloride, it is mandatory to overcome the limited scalability of the generator, by introducing a liquid target for the production of [^68^Ga]Gallium by a cyclotron. This will enable a significant reduction of costs due to the synthesis process of [^68^Ga]Ga-PSMA-11 improving the number of PET/CT exams performed at once in a similar range/fashion to [^18^F]PSMA-1007.

Our experience also shows that among the different PSMA radioligands used to image or to treat PCa, there are no significant differences in their binding within tumor lesions. As shown in Figure 1, no differences in the biodistribution in lesions were identified among patients, which has been studied before starting RLT with [^177^Lu]Lu-PSMA-617 and imaged, after two cycles of RLT, with [^68^Ga]Ga-PSMA-11. The relevant aspect is the similar distribution and levels of uptake within malignant deposits among diagnostic ligands and therapeutic companions while some differences were observed in normal organs, mainly related to greater lipophilicity of [^18^F]PSMA-1007 vs. other radioligands.

In conclusion, the choice between [^68^Ga]Ga-PSMA-11 and [^18^F]PSMA-1007 is not influenced by significant differences in the rate of detection, while it might reflect economical and/or the number of PET/CT studies to perform on a daily or weekly basis. In our analysis, [^18^F]-PSMA-1007 seems to add several advantages in routine clinical applications, related to economic convenience, greater availability, and consequent higher number of performed PET/CT exams, as well as a pattern of distribution similar to [^177^Lu]Lu-PSMA-617 within PCa lesions, which supports its clinical use for proper selection of patients eligible for RLT.

## Figures and Tables

**Figure 1 molecules-27-03862-f001:**
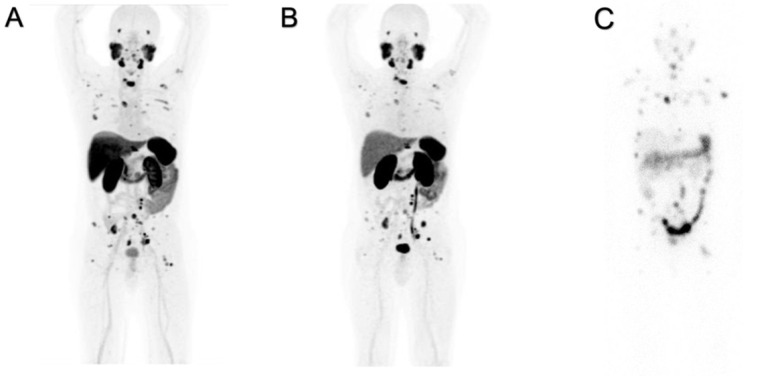
Positive PET/CT scans performed with (**A**) [^18^F]PSMA-1007 (70–90 min p.i.) and with (**B**) [^68^Ga]Ga-PSMA-11 (45–60 min p.i.) for the same patient. Planar images (**C**) acquired with SPECT/CT γ-camera 96 h after administration of [^177^Lu]Lu-PSMA-617.

**Figure 2 molecules-27-03862-f002:**
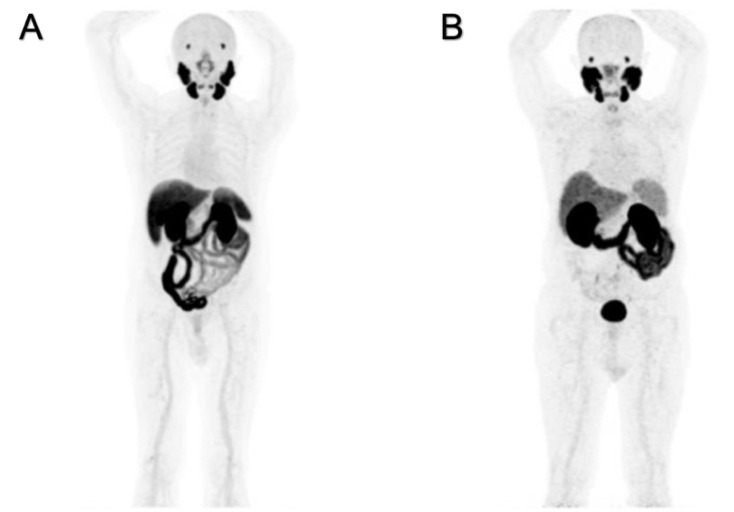
Negative PET/CT scans: patient (**A**) was examined with [^18^F]PSMA-1007, and patient (**B**) was examined with [^68^Ga]Ga-PSMA-11.

**Table 1 molecules-27-03862-t001:** Comparison of [^68^Ga]Gallium chloride starting activities (low and high) and related RCY and RCP.

[^68^Ga] Low Activity (L)			[^68^Ga] High Activity (H)		
^68^Ga Starting Activities (MBq)	RCY (%)	RCP (%)	^68^Ga Starting Activities (MBq)	RCY (%)	RCP (%)
640.84	77.70	100	1323.12	67	100
637.88	80.70	100	1536.61	71.40	99.99
572.39	83.10	100	1519.96	74.50	99.95
648.61	79.40	100	1499.98	76.10	99.91
572.39	79.20	100	1434.49	67.20	99.95
598.29	78.60	100	1448.18	72.10	99.98
582.38	78.50	100	1387.13	64.80	99.99
526.51	94.60	99.75	1365.67	74	100
580.90	77.60	100	1441.89	72.90	99.81
605.32	80.40	99	1405.63	74.80	100
**596.55 ***	**80.98 ***	**99.91 ***	**1436.27 ***	**71.48 ***	**99.96 ***
**37.97 ****	**0.05 ****	**0.01 ****	**68.68 ****	**0.04 ****	**0.01 ****

RCY (%) not corrected for decay and RCP (%) reported for every batch production of [^68^Ga]Ga-PSMA-11; for low starting activities (range L) RCY score is 80.98%, for high starting activities (range H) RCY decreased to 71.48%. * Average ** Standard deviation.

**Table 2 molecules-27-03862-t002:** Costs of production and PET/CT exams for [^18^F]PSMA-1007 and [^68^Ga]Ga-PSMA-11.

Radioligand	Costs of Production		N° PET/CT Performed		Costs for Single PET/CT Exam	
[^18^F]PSMA-1007	5.454€		25		**30€**	
[^68^Ga]Ga-PSMA-11	**L** activity	**H** activity	**L** activity	**H** activity	**L** activity	**H** activity
	1.831€	1.831€	**2**	**5**	**583€**	**275€**

Costs of each, single, production of [^18^F]PSMA-1007 and [^68^Ga]Ga-PSMA-11 correlated to the allowed number of PET/CT. The cost of each single PET/CT exam considered a standard dose of 259 MBq for [^18^F]PSMA-1007 and of 154 MBq for [^68^Ga]Ga-PSMA-11. For [^18^F]PSMA-1007 the presented range of starting activity is the most cost/effective, as previously shown [6].

## Data Availability

Raw data is available on https://zenodo.org/record/6563187#.YoY_uS98qX1, accessed on 19 May 2022.

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
