# Peer review of "Analysis of Pros and Cons in Using [68Ga]Ga-PSMA-11 and [18F]PSMA-1007: Production, Costs, and PET/CT Applications in Patients with Prostate Cancer"

_molecules, 2022, doi:10.3390/molecules27123862_

Round 1

Reviewer 1 Report

The submitted manuscript is well written and addresses an interesting topic regarding clinical PET imaging of patients with prostata cancer. Due to the availability of different suitable imaging probes it is often an important question which probe to be used. Here the submitted manuscript might give important in-put which way to even so the scientific input is rather low. As many patients are affected by this decision, the manuscript is worth publication.

However, there are some issues in the submitted manuscript which should be addressed more thoroughly.

a)                  Radiochemical Yield as function of starting radioactivity

The authors observe a decrease of radiochemical yield both for Ga-68 PSMA as well for the F-18 FPSMA. Here the Ga-68 PSMA production appears to be more radiation sensitive compared to the F-18 approach: for Ga-68 a 10 % decrease when doubling the start radioactivity from 0.7 GBq to 1.4 GBq and likewise a 10% decrease for F-18 when roughly doubling the start radioactivity from 90 to 160 GBq (data retrieved from authors previous publication).

The authors anticipate a decrease of cost per MBq for Ga-68 PSMA when larger amounts of cyclotron produced Ga-68 is employed within the synthesis. However, it is unknown for the authors how for example 5 GBq starting Ga-68 radioactivity obtained from a Zn-68 liquid target several 10th of GBq from a Zn-68 solid target will affect the radiochemical yield, ie. linear/exponential decrease of RCY or stable around 50% like observed? Therefore  the authors should either retrieve information from the literature addressing this issue or delete the statement in conclusion on page 8 line 290 – 293.

b)                  Difference in standard doses for Ga-68 PSMA and F-18 FPSMA

In table 2 the authors mention for F-18 FPSMA a standard dose for 259 MBq and 154 MBq for Ga-68 PSMA.  The authors elaborate  in the discussion page 6 line 203 – line 208 the favorable imaging properties of F-18 compared to Ga-68. In combination with the longer half-life of F-18, 109.8 min, compared to Ga-68, 68 min, one should expect actually a lower standard dose for the F-18 imaging probe, i.e. a vice versa standard dose for the F-18 and Ga-68 compound. Please explain therefore the dose regiment.

c)                   F-18 FPSMA uptake in the skeleton

A characteristic difference for F-18 FPSMA and Ga-68 PSMA is a higher number of false positive lesions in the skeleton. A likewise higher number of positive lesions is observed in bone scans performed with F-18 sodium fluoride compared to TC-99m MDP SPECT scans. For F-18 FPSMA, this is attributed to a higher photon flux density for F-18. Without further explanation, this assumption is hardly to understand for the reader. Please add therefore additional information how photon flux density results in the detection of more false positive lesions.

d)                  Contradiction in Discussion page 7 line 219 – 225 and Conclusion page 8 line 305 – 310

In the discussion the authors state that FPSMA shows a higher uptake in recurrent metastases compared to Ga-68 PSMA. Furthermore a more similar biodistribution for F-18 FPSMA and Lu-177 PSMA-617 is described. In combination with the superior imaging properties of F-18 compared to Ga-68, FPSMA seems to be a superior imaging probe compared to Ga-68 PSMA. However, in the conclusion line 305-206 the main statement is that the use of F-18 FPSMA or Ga-68 PSMA may not be influenced by significant difference in the rate of detection.

Either F-18 FPSMA is better and then this should be stated as well in the conclusion. If the superior imaging properties of F-18 FPSMA does not matter in regular clinical use, than this should be stated and explained in the discussion.

e)                  Other

Discussion line 158 and 159: mCi used in stead of MBq. Please change to SI unit MBq.

Conclusion page 8 line 299: No these three radiopharmaceuticals. Fragment, please delete.

  Author Response

a)Radiochemical Yield as function of starting radioactivity

Unfortunately we don’t have data relatively the behavior of the RCY with different starting activities of [68Ga]Gallium, although the amount of final product [68Ga]Ga-PSMA-11 seems to be always higher than the amount obtained with generator and the quality of 68Ga-labeled peptide is unaffected.

In lines 206-211 we gave explanations and the related references, so we didn’t modify the conclusions.

b)Difference in standard doses for Ga-68 PSMA and F-18 FPSMA

[18F]Fluoride allows to have a better spatial resolution considering the administered activity corrected for body weight. According to standard guidelines, also considering pharmacokinetics properties and the timing for PET/CT acquisition after injection, for a patient with a weight of 70kg, the administered activity is 2,2 MBq/kg for 68Ga (154 MBq) and 3,7 MBq/kg for 18F (259 MBq).

c)F-18 FPSMA uptake in the skeleton

With the expression “photon flux” we referred to the major photon emission density of [18F] than [68Ga].

In lines 259-264 we gave more explanations.

d)Contradiction in Discussion page 7 line 219 – 225 and Conclusion page 8 line 305 – 310

In lines 247-251 we gave more explanations to make clear the object in discussion. In fact, in the conclusions (308-314) we highlighted all the advantages of [18F]-PSMA-1007 that make it the better probe for the routinary clinical applications, in particular economical aspects, major availability, the higher number of performed PET/CT exams. Moreover from our experience  we observed that there is an higher uptake within lesions with [18F]-PSMA-1007 while the overall rate of detection is not significantly affected by the used PSMA-probe.

e) other

Discussion line 158 and 159: mCi used in stead of MBq. Corrected

Conclusion page 8 line 299: No these three radiopharmaceuticals. Fragment removed.

Reviewer 2 Report

In the manuscript entitled “Analysis of Pros and Cons in using [68Ga]Ga-PSMA-11 and [18F]PSMA-1007: production, costs and PET/CT applications in patients with Prostate Cancer” the two diagnostic radioligands are compared and their disadvantages and advantages regarding  for example costs and time of radiolabeling procedures are described.

 Comments/corrections: Ph. Eur. XII

-          Abstract: A short sentence should also be added about the costs as well.

-          Line 18/19: remove the third digit after the point – only two digits after the point should be used.

-          Line 56: [68Ga]-labeled and [18F]-labeled is wrong – change to 68Ga-labeled and 18F-labeled

-          Line 57: change “our experience” to “our data”

-          Line 65: remove “the” before [18F]PSMA-1007

-          Line 71/72: The authors should add the number of labellings carried out (n=10).

-          Line 88: remove “respectively”

-          Line 88: remove the point after “H”

-          Line 89: add a comma before “respectively”

-          Line 86/87/88: Two digits after the comma/point are sufficient – authors should remove the third digit from ±0.051%, ±0.038%, ±0.001% and 0.002%.

-          Line 94: Table 1: The authors should be consistent with writing digits after the comma/point – all numbers should have two digits after the comma.

-          Line 100: Where do they 37 MBq come from – why the authors use this amount (1 mCi?)? Authors should clarify that point.

-          Line 101: This sentence is not clear – aren’t the number of PET/CT exams dependent on the available activity of 68Ga-PSMA? Please clarify.

-          Line 101: add a comma before “respectively”

-          Table 2: change [18Ga] to [68Ga]

-          Table 2: Remove 583€ and 275€ in the second line

-          Line 114: change “coist” to “cost”

-          Line 136: change “ones” to “SPECT/CT images”

-          Line 148: change “follow up” to “follow-up”

-          Line 151/152: change to 68Ga-labeled and 18F-labeled

-          Line 157: change to 68Ga-labeled

-          Line 158/159: The authors use Bq throughout the manuscript so they should change “mCi” to Bq (MBq).

-          Line 183: change “Institutions” to “institutions”

-          Line 208: change “pet” to “PET”

-          Line 213: Figure 2 – please clarify if this is the same patient or not.

-          Line 231: Please add Germany after Eckert-Ziegler

-          Line 230: remove “Germanium Chloride” and change to “68Ga/68Ge generator”

-          Line 258: change “by Pharmacopoeia” to “in Ph. Eur. XII”

-          Line 279: change to “…center where only PET/CT is available.”

-          Line 284: change “results the” to “results to be the”

-          Line 290: change “is mandatory” to “it is mandatory”

-          Line 291: change “of generator” to “of the generator”

-          Line 296: change “prostate cancer” to “PCa”

-          Line 296: Correct sentence to: “…among a patient which has been studied before starting…”

-          Line 299: remove “No these three radiopharmaceuticals”

-          Line 309: change “resulted in our” to “based on our”

-     Line 310: change “PC” to “PCa”

Author Response

We corrected all the reported errors